



**Multispecies and high spatiotemporal resolution database of vehicular**
**emissions in Brazil**
Leonardo Hoinaski[a,*], Thiago Vieira Vasques[b], Camilo Bastos Ribeiro[b] and Bianca Meotti[b]
[a]Department of Sanitary and Environmental Engineering, Federal University of Santa Catarina, Flo-
rianópolis, Santa Catarina, Brazil, leonardo.hoinaski@ufsc.br
[b]Graduate Program in Environmental Engineering, Federal University of Santa Catarina,
Florianópolis, Santa Catarina, Brazil, vvthiago@hotmail.com, cb_ambiental@hotmail.com,
meottibianca@gmail.com
*Corresponding author
*E-mail address:* leonardo.hoinaski@ufsc.br (L. Hoinaski).





**Abstract** – In this article, we present the BRAzilian Vehicular Emissions inventory Software
(BRAVES) database, a multispecies and high spatiotemporal resolution database of vehicular emis-
sions in Brazil. We provide this database using spatial disaggregation based on road density, temporal
disaggregation using vehicular flow profiles, and chemical speciation based on SPECIATE database
from the United States Environmental Protection Agency. Our BRAVES database provides hourly
and annual emissions of 41 gaseous and particle pollutants. Users can define the spatial resolution,
which ranges from a coarse to very refined scale Spatial correlation analysis reveals that the BRAVES
database reaches similar performance to the vehicular emissions inventory from Emissions Database
for Global Atmospheric Research (EDGAR). A comparison with Modern-Era Retrospective analysis
for Research and Applications, Version 2 (MERRA-2) surface concentration confirms the con-
sistency and reliability of the BRAVES database on representing the spatial pattern of vehicular emis-
sions. Compared to EDGAR, the BRAVES database brings more spatial, temporal, and chemical
details. These additional features are crucial to understanding important atmospheric chemistry pro-
cesses in Brazil. All codes and inputs are freely available, and the outputs are compatible with the
input requirements of sophisticated chemical transport models. We envision that our database will
enable the scientific and environmental community to gain new insights into vehicular emissions and
their effects in Brazil, where emissions inventories are scarce and urgently needed.

**Keywords:** Vehicular emissions, temporal disaggregation, chemical speciation, spatial disaggrega-
tion, Brazil.





## 1. Introduction


Vehicular emissions threaten urban air quality (Brito et al. 2018; Sawyer, 2010) and cause
several environmental damages from local to global scales. These emissions deleteriously affect hu-
man health (Anenberg et al. 2017; Krzyzanowski et al. 2005) and contribute to the increase in the
concentration of greenhouse gases in the atmosphere (Shindell et al. 2011; Unger et al. 2010).
It is challenging to control vehicular emissions in developing countries where city growth is
disorganized manner and vehicle population increases dramatically (Lyu et al. 2020; Sun et al. 2020).
Furthermore, vehicular emissions inventories, an essential tool to control air pollution, are scarce and
limited to wealthy cities (Huneeus et al. 2020) and developed countries. When available, an emissions
inventory often does not provide the required data to design air quality management systems.
Brazil has experienced a rapid rise in its vehicular fleet (Carvalho et al. 2015) and transport
volume. Even though the program to control vehicular emissions has reduced the emissions from the
transport sector in Brazil (Andrade et al. 2017), vehicles are still potentially the dominant source of
air pollution in several municipalities.
The impact of vehicular emissions in many Brazilian municipalities is still unknown (Ribeiro
et al. 2021). Current inventories provide only annual emissions from national to municipality scales,
not reaching the spatial and temporal resolution necessary for air quality modeling (Álamos et al.
2022), nor the concentration of chemical species that participate in chemical reactions in the atmos-
phere. For this reason, most regional air quality assessments in Brazil rely on global emissions inven-
tories, which have been proved to be biased against local inventories. Also, global inventories do not
present enough spatial and temporal resolution for regional and local studies (Ibarra-Espinosa et al.
2018). Even in the megacity of São Paulo, where the air quality network is well developed and mul-
tiple inventories have been developed, there is still room for improvement in emissions inventories
(Andrade et al. 2017), especially regarding chemical species involved in the photochemical process
in the atmosphere.



In this article, we present the first comprehensive multispecies high spatiotemporal resolution
database of vehicular emissions for the entire Brazilian territory. The BRAzilian Vehicular Emissions
inventory Software (BRAVES) database has spatial disaggregation based on road density, temporal
disaggregation using vehicular flow profiles, and chemical speciation from the US EPA SPECIATE
database. The BRAVES database provides hourly and annual emissions of 41 gaseous and particle
pollutants. Users can define the spatial resolution from coarse to very refined scales. The emissions
are derived from the BRAVES model (Vasques and Hoinaski, 2021), which uses a probabilistic ap-
proach that accounts for the fleet characteristics, fuel consumption, vehicle deterioration, and inten-
sity of use, to calculate the vehicular emissions from the exhaust, tires, roads, brake wear, soil resus-
pension, refueling, and evaporative emissions. Here, we present methods and a comparison between
the BRAVES database and independent databases. We also make all codes and inputs freely availa-
ble.



## 2. Vehicular emissions data

Our database uses output data from the BRAzilian Vehicular Emissions inventory Software – BRAVES (Vasques and Hoinaski, 2021), which employs a probabilistic bottom-up method to estimate vehicular emissions aggregated by municipality (Figure 1). BRAVES estimates the vehicular emissions from the exhaust, tires, roads, brake wear, soil resuspension, refueling, and evaporative emissions. The software provides, by fleet category (i.e., commercial-light vehicles, motorcycles, light-duty and heavy-duty vehicles), annual emissions of carbon monoxide (CO), carbon dioxide ($CO_2$), methane ($CH_4$), hydrocarbons (HC), aldehydes (RCHO), non-methane volatile organic compounds (NMVOC), nitrogen oxides (NOx), particulate material (PM), nitrous oxide ($N_2O$), and sulfur oxide ($SO_2$). Throughout this article we call current database the BRAVES database. Codes and outputs from BRAVES are available by registering at https://hoinaski.prof.ufsc.br/BRAVES/ and https://github.com/leohoinaski/BRAVES, where users can access instructions to run the database and download the input files. The outputs are generated in netCDF format and with annual or hourly resolutions.

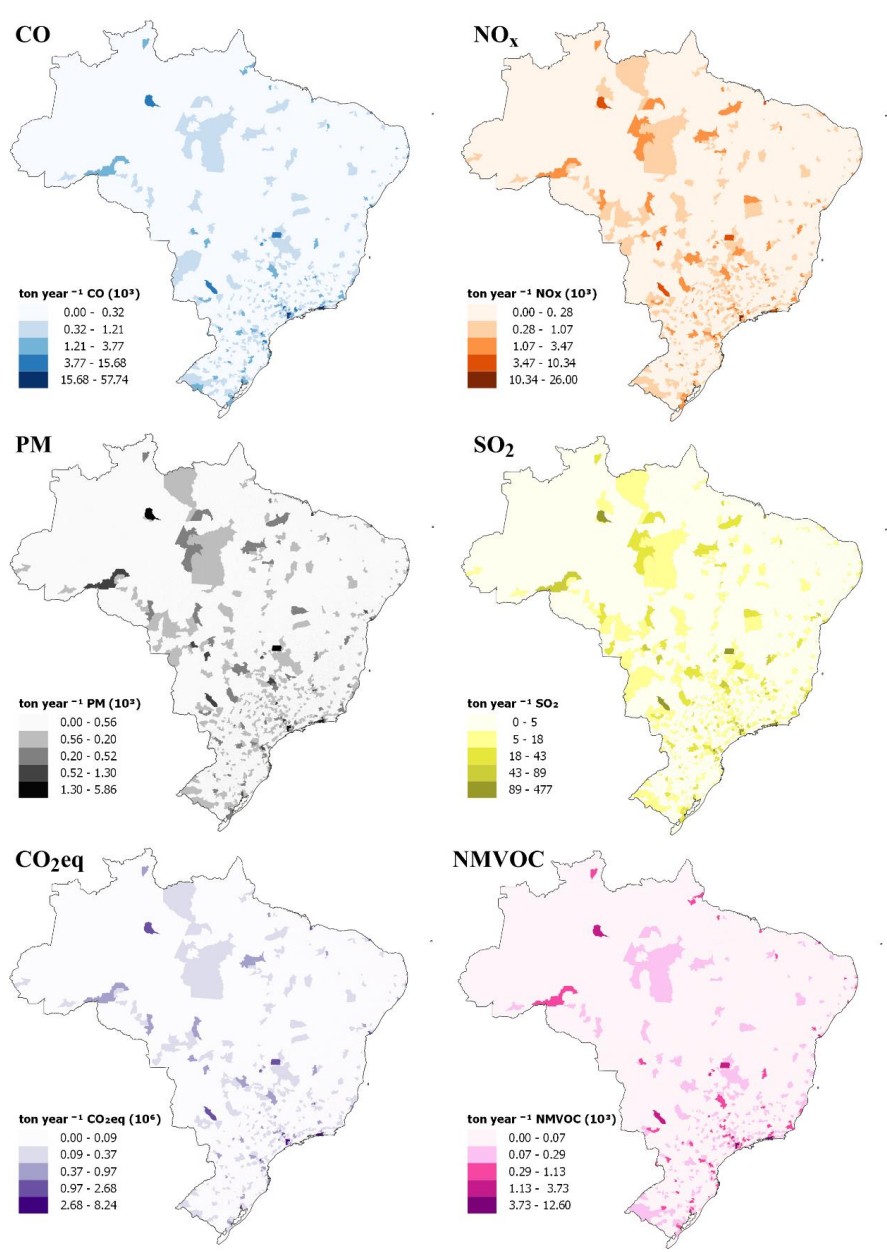


**Figure 1.** Vehicular emissions inventory in 2019 of a) CO, b) NOx, c) PM, d) SO$_2$, e) CO$_2$, and f)

NMVOC provided by BRAVES (Vasques and Hoinaski, 2021).


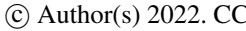

**3. Spatial disaggregation**

Since vehicular emissions from BRAVES are aggregated by municipality, we use a road den-
sity approach to distribute the emissions of each municipality in pixels with user-defined resolution.
Previous work of Tuia et al. (2007) and Gomez et al. (2018) shows that road density is one of the
most reliable approaches to disaggregate vehicular emissions. In this article, road density factor
($RD_{p,m}$) is calculated by the sum of road lengths ($L_p$) on each pixel ($p$) divided by the total road
length ($L_m$) inside a municipality ($m$) (Equation 1). Road shapefile data derived from OpenStreetMap
(www.openstreetmap.org) can be downloaded at (https://download.geofabrik.de/south-america/bra-
zil.html) for Brazilian territory. Figure 2 shows the spatial distribution of $RD_{p,m}$ in Brazil. Multiplying
$RD_{p,m}$ by the vehicular emissions in each municipality derived from BRAVES provides the spatial-
ized emission ($E_{p,m,c}$) of compound $c$ in pixel $p$ within a municipality $m$ (Equation 2). We provide a
parallelized method to estimate the road density in Brazil at https://github.com/leohoinaski/BRAVES.

$RD_{p,m} = \frac{L_p}{L_m}$                             (Eq.1)

$E_{c,p,m} = RD_{p,m} \times E_{c,m}$                     (Eq.2)

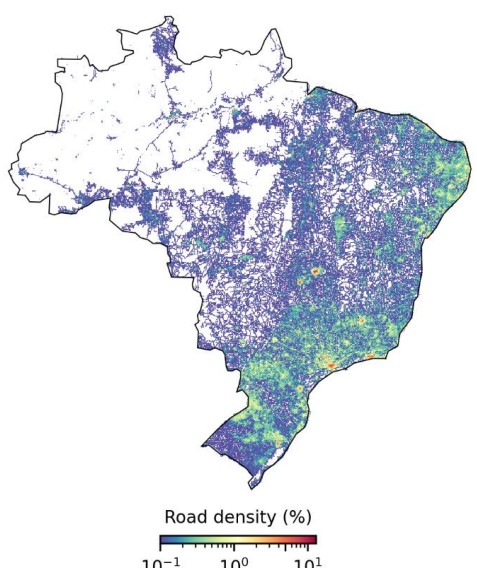

**Figure 2.** Road density factor in Brazil.

Figure 3 shows the annual emissions of aldehydes and CO from 2019 in Brazil by fleet category using spatial distribution based on the road density approach. Hotspots of vehicular emissions are concentrated in urbanized areas in Brazil (Figure 3). Among fleet categories, heavy-duty is the major emitter of aldehydes, while light-duty emits the most CO. Vasques and Hoinaski (2021) presents a full comparison of vehicular emissions from each fleet category in Brazil using BRAVES. Figures SM1 to SM5 demonstrate the BRAVES database by Brazilian state.

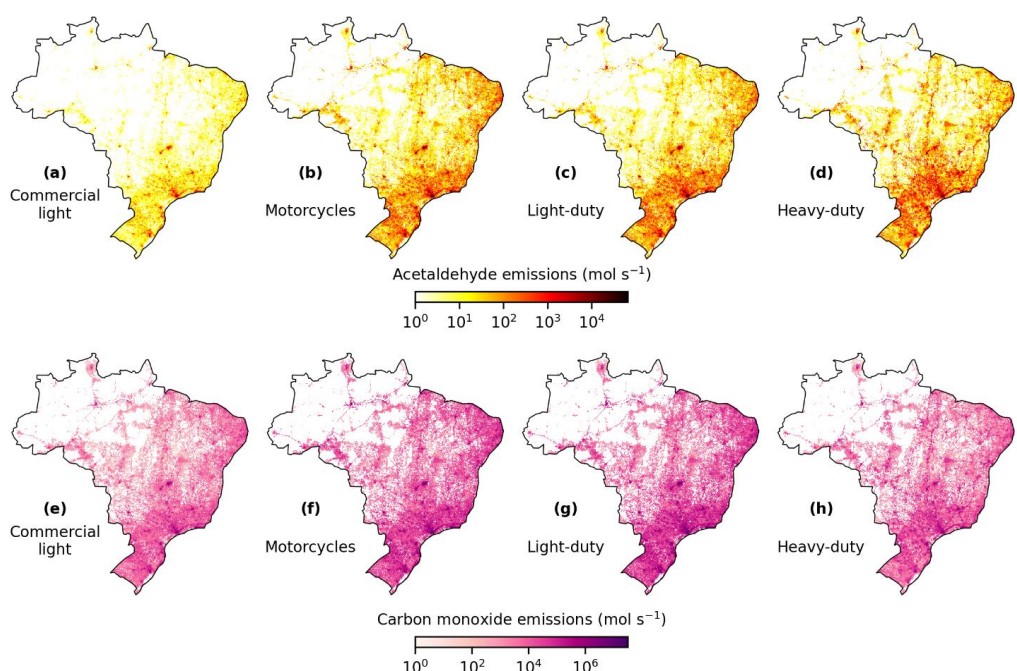


**Figure 3.** Vehicular emissions of aldehydes a-d) and carbon monoxide e-h) from commercial light,

126                    motorcycle, light, and heavy-duty fleets in 2019 in Brazil.


**4. Temporal disaggregation**

130          Temporal disaggregation based on traffic flow observations from Environment and Water Re-

sources Institute from Espírito Santo state (IEMA ES, 2019) splits original annual emissions from
BRAVES into hourly basis emissions. In this article, the temporal disaggregation factor (Figure 4) is
composed of hourly, weekly, and monthly traffic factors. Hourly emissions ($E_{c,p,m,h}$) of each air pol-
lutant are obtained by the multiplication of $E_{p,m,c}$ and the temporal disaggregation factor ($T_f$) (Equa-
tion 3).
$E_{c,p,m,h} = T_f \times E_{c,p,m}$                                                      (Eq.3)



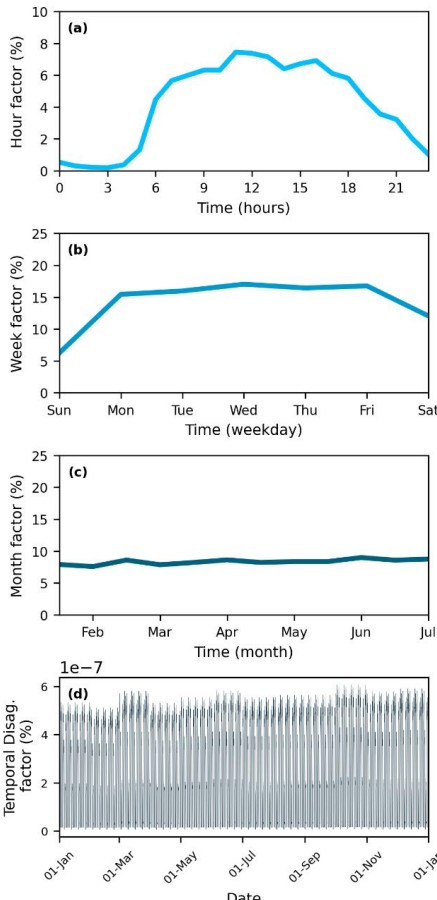


**Figure 4.** Temporal disaggregation factor and its components a) Hourly factor, b) Weekly factor, c)
Monthly factor, d) Annual to hourly basis temporal disaggregation factor.



## 5. Chemical speciation



We use data from SPECIATE 5.1 (US EPA, 2020; Eyth et al. 2020) from the United States
Environmental Protection Agency - US EPA (https://www.epa.gov/air-emissions-modeling/speciate)
to speciate emissions of chemical constituents of Volatile Organic Compounds (VOC) and PM, which
has not been previously estimated by BRAVES. The chemical speciation method also converts NOx
to NO and $NO_2$.
Regarding the speciation procedures, in this article we group light duty and commercial light
vehicles, and motorcycles together as light vehicles. We select profiles to speciate PM emissions from
the exhaust of heavy and light vehicles, soil resuspension (road dust), tire wear, and brake wear. VOC
emissions from the exhaust and evaporative process are also speciated. Table SM6 in the supplemen-
tary material summarizes the profiles from SPECIATE 5.1 used in the chemical speciation. We target
the species required for Carbon Bond Chemical Mechanism (Yarwood et al. 2010ab) version 6
(CB06), which describe tropospheric oxidant chemistry in a concise manner suitable for use in com-
plex 3-dimensional atmospheric models (Yarwood et al. 2010ab). Table 1 presents the VOC and PM
compounds considered in the chemical speciation in this database.





158    Table 1. List of chemical species included in the BRAVES database (Yarwood et al. 2010b).

| Specie | ID | Specie | ID |
|---|---|---|---|
| Acetone | ACET | Aluminum | PAL |
| total Acrolein | ACROLEIN | Calcium ion | PCA |
| total Acetaldehyde | ALD2 | Chloride ion | PCL |
| Benzene | BENZ | Elemental carbon | PEC |
| 1,3-Butadiene | BUTA13 | Iron | PFE |
| Ethanol | ETH | Potassium ion | PK |
| Ethane | ETHA | Magnesium ion | PMG |
| Ethyne | ETHY | Manganese | PMN |
| Formaldehyde | FORM | Sodium ion | PNA |
| Isoprene | ISO | Ammonium | PNH4 |
| Naphthalene | NAPH | Nitrate | PNO3 |
| Propane | PRPA | Silicon | PSI |
| Monoterpenes | TERP | Sulfate | PSO4 |
| Toluene | TOL | Titanium | PTI |
| xylene | XYLMN | | |


160    The chemical speciation factors employed to split VOC and PM emissions are calculated by

161 the average of the weighting percentage of the corresponding species from SPECIATE 5.1. We con-

162 sider exhaust, evaporative, and particulate emissions of light and heavy vehicles. Figures 5 and 6

163 show the speciation factor used to generate the database. Multiplication factors of 0.495 and 0.505

164 derived from SPECIATE 5.1 convert NOx emissions to NO and $NO_2$, respectively. Table SM7 sum-

165 marizes the speciation factors used to build this database.


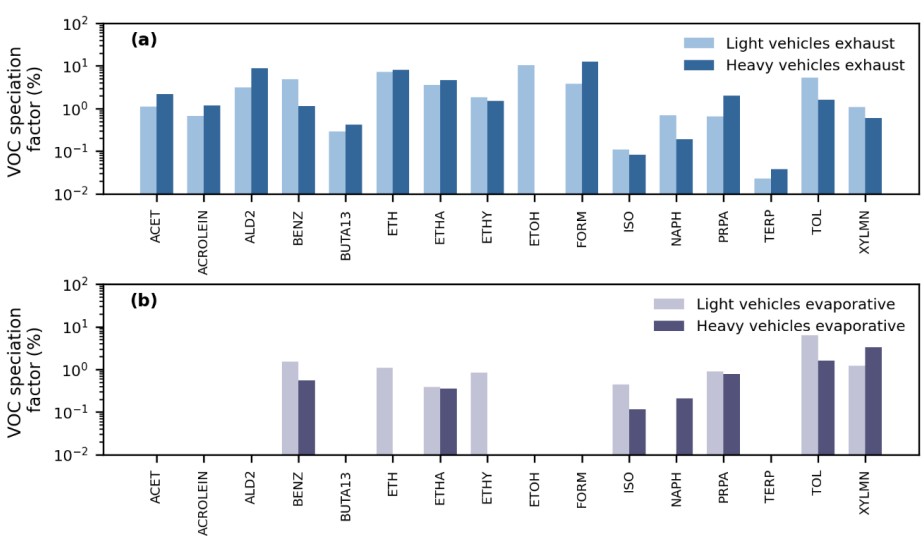

**Figure 5.** VOC chemical speciation factor for exhaust and evaporative emissions from light and

heavy vehicles used in this article.

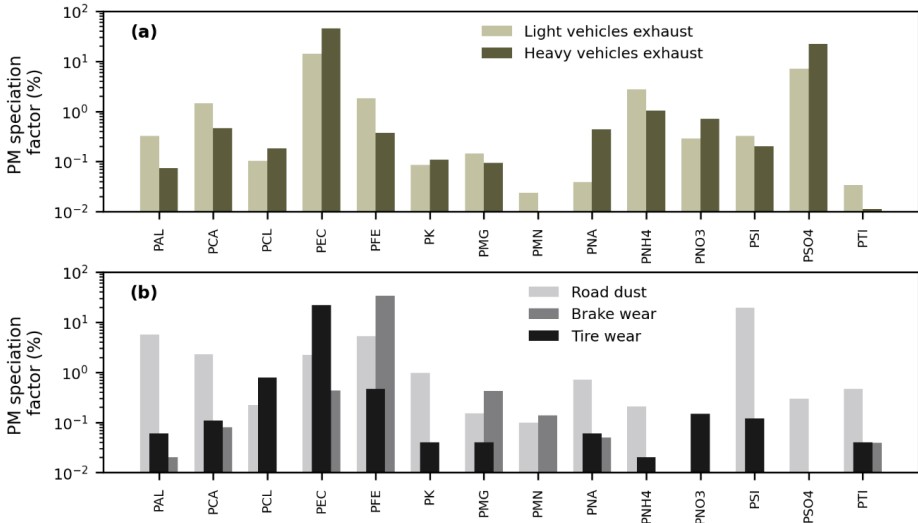


**Figure 6.** PM chemical speciation factors a) exhaust emissions from light and heavy vehicles b)

road dust resuspension, brake wear, and tire wear.






**6. The database and codes**



The database contains hourly emissions of 41 chemical species, such as ACET, ACROLEIN,
ALD2, BENZ, BUTADIENE13, $CH_4$, CO, $CO_2$, ETH, ETHA, ETHY, ETOH, FORM, ISO, $N_2O$,
NAPH, NO, $NO_2$, PAL, PCA, PCL, PEC, PFE, PK, coarse mode primary PM (PMC), PMG, PMN,
unspeciated $PM_{2.5}$ (PMOTHR), PNA, $PNH_4$, $PNO_3$, POC, PRPA, PSI, $PSO_4$, PTI, $SO_2$, TERP, TOL,
VOC, and XYLMN. We provide a code to generate hourly resolved files with a user-defined grid for
a single or whole group of species (https://github.com/leohoinaski/BRAVES). These files are com-
patible with the input requirements of sophisticated chemical transport models, such as the Commu-
nity Multiscale Air Quality Model (CMAQ), the Weather Research and Forecasting (WRF) model
coupled with Chemistry (WRF-Chem), the Comprehensive Air Quality Model with Extensions
(CMAx), and others. Smaller domains and finer resolution can be easily created by modifying in the
python codes. Figure 7 shows the vehicular emissions of Benzene in Brazil on January 1$^{st}$, 2019 using
the BRAVES database.

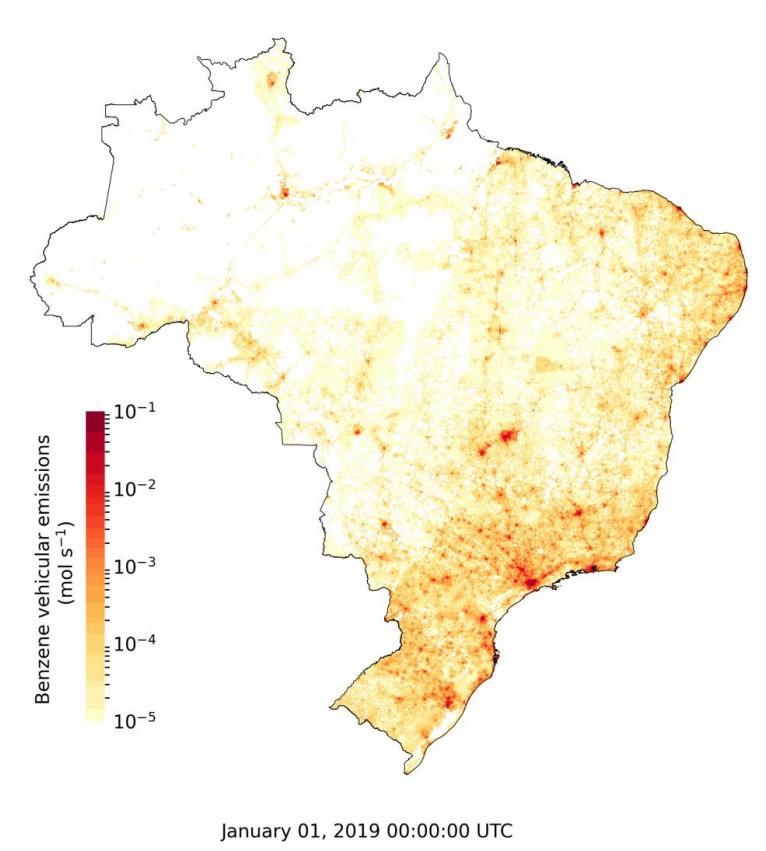

January 01, 2019 00:00:00 UTC


**Figure 7.** Vehicular emissions of Benzene in Brazil on January 1st, 2019 using the BRAVES data-

base.

We provide the BRAVES database annual speciated emissions with 0.05°x0.05° of resolution

covering the entire Brazilian territory at: https://doi.org/10.5281/zenodo.6141109





**7. Comparison with independent databases**

We analyze the spatial correlation and bias between the BRAVES database and the Emissions
Database for Global Atmospheric Research (EDGAR – version 5.0 https://edgar.jrc.ec.europa.eu/da-
taset_ap50) annual gridmaps (Crippa et al. 2018; European Comission 2022). We performed the com-
parison using the "Road Transportation" emissions from EDGAR for the Brazilian territory, includ-
ing soil resuspension emission rates of $PM_{10}$ from EDGAR. The BRAVES database emission rates
in tons per year from 2015 were regridded to the same spatial resolution of EDGAR. The Spearman
coefficient estimates the spatial correlation, while the difference in absolute emissions calculates the
bias between the datasets. We compare the disaggregated emissions of CO, $PM_{10}$, NOx, and COV
from BRAVES and EDGAR.

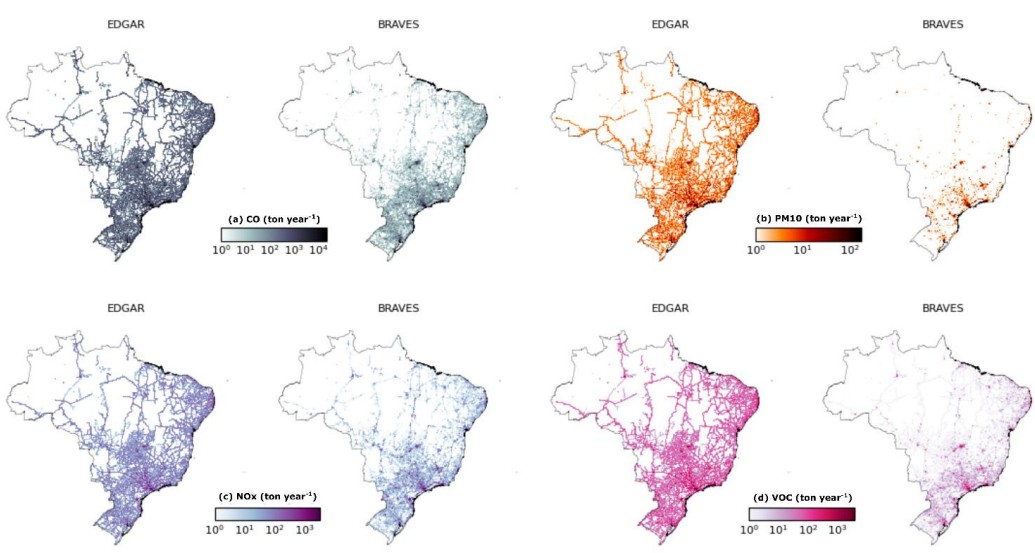


**Figure 8**. Comparison of a) CO, b) $PM_{10}$, c) NOx, and d) VOC spatial distribution (log scale) pro-

209                          vided by the EDGAR and BRAVES databases.




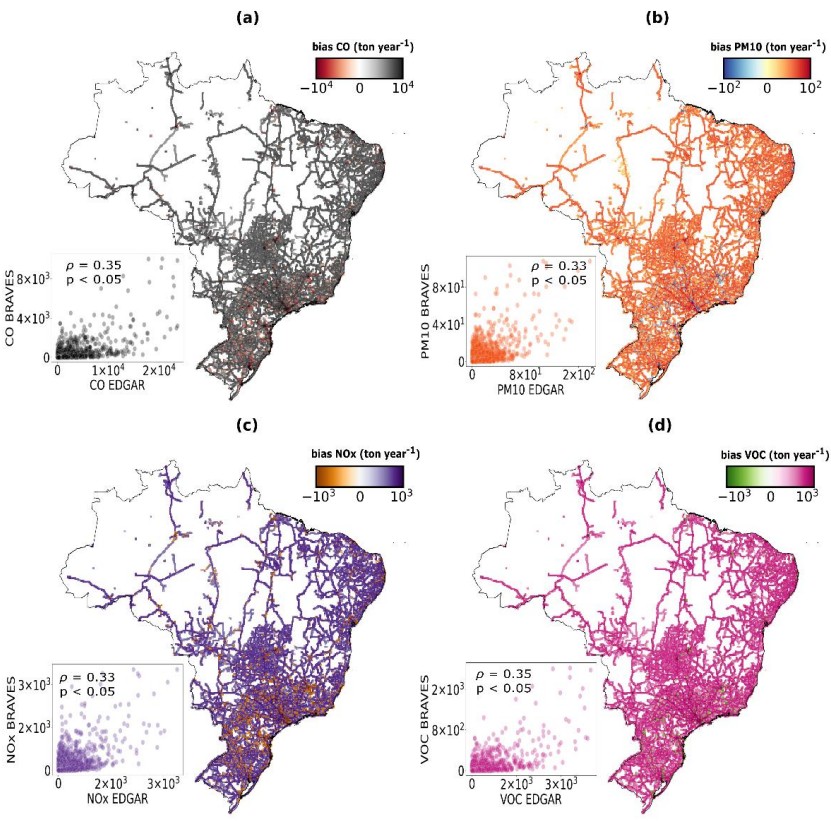


**Figure 9**. Bias (log scale) and scatter plots of a) CO, b) PM$_{10}$, c) NOx, and d) VOC emission rates

provided by the BRAVES database and EDGAR in 2015.


Emission from BRAVES and EDGAR presents overall spatial correlation ($p<0.05$) of $\rho = 0.35$

for CO, $\rho = 0.33$ for PM$_{10}$, $\rho = 0.33$ for NOx, and $\rho = 0.35$ for VOC (Figure 9). Emissions from

EDGAR are consistently higher (Figure 9) than from BRAVES, as also reported by Vasques and

Hoinaski (2021). The largest differences are observed in CO emissions, followed by VOC, NOx, and

PM$_{10}$. Madrazo et al. (2018) explains that most of the road transport Emission Factors are

overestimated in EDGAR, while Huneeus et al. (2020) found discrepancies between EDGAR and



local/national city emissions. Álamos et al. (2022) also reported an overestimation of EDGAR
emissions.
We analyze the spatial correlation between vehicular emissions of CO from the BRAVES
database, EDGAR, and surface concentration of CO from the Modern-Era Retrospective Analysis for
Research and Applications - MERRA-2. The Global Modeling and Assimilation Office (GMAO),
managed by NASA, provides MERRA-2 reanalysis products in a spatial resolution of 0.625°×0.5°,
covering from 1980 to the present (Gelaro et al. 2017; Randles et al., 2017). We calculate the annual
average concentration in 2015 from monthly data in netCDF files available at GES-DISC platform
(https://disc.gsfc.nasa.gov/datasets/M2TMNXCHM_5.12.4/summary). All grids are realigned to
match the MERRA-2 spatial resolution. We analyze the spatial correlation by Brazilian state since
the vehicular emission has more influence in urbanized ones. We assume that those cells which have
the vehicular emissions as the major source of air pollutant also have higher surface concentration.
However, this assumption has several limitations and should be carefully evaluated since it does not
account for the dispersion process and other source types (i.e., industrial, biomass burning, biogenic
sources).
Figure 10 shows the (a) CO concentrations from MERRA-2 in São Paulo (SP) state, (b) ve-
hicular emission of CO from EDGAR, and (c) vehicular emission of CO from the BRAVES database.
We highlight the state of SP since it has approximately 7 million vehicles, being considered the state
with the highest vehicular emission in Brazil (Vasques and Hoinaski, 2021). The BRAVES database
and EDGAR reach similar spatial correlation with MERRA-2 (Figure 10). However, the zoom-in
quadrant in the São Paulo metropolitan region reveals the greater level of details from the BRAVES
database compared to EDGAR. In addition, BRAVES has higher temporal resolution and chemical
speciated emissions. In other Brazilian states, such as Minas Gerais (MG) and Rio Grande do Sul
(RS), there is also a positive correlation between vehicle emissions and surface concentrations of CO



(Table SM8). It shows that both databases consistently capture he spatial variability of vehicular
emissions and the BRAVES database brings additional features for air quality studies in Brazil.

**Figure 10**. CO spatial distribution provided by (a) MERRA2, (b) EDGAR, and (c) BRAVES. Scatter plots of CO vehicle emission and CO surface concentrations in SP.



## 8. Data availability


The BRAVES database is freely available at https://doi.org/10.5281/zenodo.6141109
(Hoinaski et al., 2022). We provide annual speciated emissions with 0.05°x0.05° of resolution cov-
ering the entire Brazilian territory. Codes to generate the database are available at:
https://github.com/leohoinaski/BRAVES. Using the annual files, users can derive hourly basis emis-
sions through the available codes.



## 9. Conclusions


Here, we introduce the BRAVES database, the first high-resolution and chemical speciated
database of vehicular emissions covering the entire Brazilian territory. The BRAVES database con-
tains emissions of 41 air pollutants, from annual to hourly basis temporal resolution and user-defined
spatial resolution. The attributes of this emission database are fully compatible with sophisticated air
quality models. Moreover, the emissions of multiple chemical species presented here provide essen-
tial information to understand important atmospheric chemistry processes in Brazil. We also provide
python scripts for users who want to create their custom gridded inventory.
Even though detailed emission inventories are required to control air pollution, vehicular
emissions are scarce in most developing countries. So far, Brazil has lacked a comprehensive and
easily accessible database of vehicular emissions, and, creating gridded inventories in South America
is urgently needed. This work contributes to overcoming this gap.
The spatial correlation analysis reveals that the BRAVES database agrees with the vehicular
emissions from EDGAR, even though EDGAR emissions are consistently higher than those
BRAVES ones. We conclude that this database can be a better alternative to represent the spatial
variability of vehicular emissions in Brazil. The BRAVES database has a similar performance repre-
senting the spatial pattern of vehicular emissions, with more spatial, temporal, and chemical details
when compared with EDGAR. Moreover, the BRAVES database is in closer agreement to local and
very detailed emissions inventories. A comparison with MERRA-2 surface concentration confirms
the consistency of the BRAVES database.
Even though the present database is a step forward for air pollution research in Brazil, there
are several opportunities for expanding and improving this work. Most heavy-duty emissions occur
in high flow and high-speed limit roads, such as expressways. Future versions could improve the
spatial disaggregation in pixels containing roads with high traffic flow and/or high-speed traffic
through the optimization of the disaggregation factors. Different criteria for light and heavy vehicles



would also be needed. Moreover, the chemical speciation could include profiles to consider the Bra-
zilian reality as biofuels, fleet motorization, and regionalized soil resuspension properties. Temporal
variability would also be improved by regionalizing the profiles to account for the traffic flow in each
location.





**Acknowledgments**



288         Authors would like to thank the Fundação de Amparo à Pesquisa e Inovação de Santa Catarina

- FAPESC, for financial support of project number 2018TR499.



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

ports/pm/5820784005FY1026-20100922-environ-cb6.pdf