# Peer review of "Multispecies and high spatiotemporal resolution database of vehicular emissions in Brazil"

_Earth System Science Data, 2022_

## Author Comment (AC3)

**Multispecies and high spatiotemporal resolution database of vehicular emissions in Brazil**

MS No.: essd-2022-74

Dear reviewers,

We greatly appreciate the comments and suggestions you provided, which are very constructive and have contributed to enhance the content of the revised manuscript. Please find below the reply to all the reviewer's comments.

Best regards

Leonardo Hoinaski and coauthors

**Reply to comments by Reviewer #1, Dr. Sergio Ibarra:**

Reply to major comments

It is important to check the validity of comparison of the emissions with MERRA. The thing is that, yes, it is possible to compare emissions with concentrations but under specific conditions, for instance, when the boundary layer is low. In this way, the air pollutant concentrations should be representative of the emissions, as shown by Gallardo et al., 2012. As the database provided by Hoinaski et al present hourly factors, my recommendation is to compare under a similar set of conditions such as the ones presented by Gallardo et al. Author can find similar research on literature. Regarding the air pollutant concentrations, authors could use the CAMS global reanalysis (EAC4) Copernicus https://www.ecmwf.int/en/forecasts/dataset/cams-global-reanalysis.

Reply:   We appreciate your suggestions. We have improved this discussion and reinforced the limitations when using this approach. Indeed, a comparison between datasets in specific regions and hourly averaging time would be more accurate. We have included a comparison according to your suggestion. To compare EDGAR, BRAVES database, and MERRA on an hourly basis, we have multiplied both BRAVES database and EDGAR annual emissions by a temporal disaggregation factor. We will analyze these three databases on 01/01/2013 at 8:00, when the boundary layer is not fully developed, and the traffic has the first peak period.

Lines 55-58: "Current inventories provide only annual emissions not reaching the spatial … resolution… nor the concentration of chemical species…" According the to these words, the authors are stating the one problem of the emissions inventories are not providing concentrations. This is conceptually wrong because emissions are mass and concentrations mass over volume. I think the author meant the inventories usually do not provide the required speciation, which would be an English problem, but need to check.

Reply: Yes, you are right. We have fixed this issue in the manuscript by replacing "nor the concentration of chemical species" with "nor the emission of chemical species".

Line 85: According to the CETESB (2019, the same reference used by author), RCHO is actually the sum of aldehydes and formaldehyde. Furthermore, CETESB also provides NMHC - ETOH emission factors. In addition, as the road transportation fuel consumed in Brazil has a vastly origin on bio-fuels, 27% of gasoline is ethanol and 7% of diesel is bio-diesel, this results in a unique chemical signature of the chemical composition of fuel, hence the emissions. Actually, there are literature mentioning the high number of carbonyls in the vehicular emissions (Nogueira et al., 2015). According to table SM7, the species C2H4O (ALD2), C2H6O (ETOH) and CH2O (FORM PRIMARY) are present in the NMHC speciation. Then, in order to provide a correct speciation, the author must reply the following questions:

Which pollutant are they using to perform the speciation? NMHC or NMHC – ETOH? Do the NMHC emission factors already consider RCHO? If the authors are using NMHC, the it supposed that ETOH and RCHO is already part of NMHC. However, this would result in a different chemical composition signature. Then, the recommendation would be preserving the proportions of ALD, FORM and ETOH and recalculate the speciation for the other compounds. I would say that this is more important for ETOH than RCHO. Can the author comment on that? Is this part of an ongoing work?

Reply: Thank you for this very insightful comment. We have fully adapted the code to preserve the original estimates using CETESB's emission factors, which is better than the speciation factors from Speciate to represent the ETOH and RCHO emissions in Brazil.

The estimates of total aldehydes (ALDX) are based exclusively on CETESB emission factors. Now, we speciate the acetaldehydes (ALD2 and ALD2_PRIMARY) and formaldehyde (FORM and FORM PRIMARY) using data from local studies. We have considered that aldehydes represent 50% and 22% of ALDX, from light-duty and heavy-duty vehicles respectively. Formaldehyde emissions (FORM and FORM_PRIMARY) represents 39% (light-duty) and 69% (heavy-duty) of ALDX emissions

We have modified the code and database to include the ETOH emissions from Flex Fuel vehicles running with ethanol when CETESB's emission factor is available (since 2018). ETOH emissions from other vehicles will still be estimated using NMHC speciation factors from Speciate.

References:

DOI: 10.1016/j.fuel.2014.05.091

https://fapesp.br/eventos/2016/02/mc/Maria_de_Fatima.pdf

Lines 102-104: While this method is conservative, the emissions need to be considered are currently expressed as points in space and not mass flux. Then, the current format is not appropriate for air quality models. In other words, the temporal mass must be divided by the area, for instance, the gases inputs to the WRF Chem model are expressed as $\mu g/km^2/h$. Then, the authors must recalculate the NetCDF outptus or add the proper flag indicating that the user must do divide by the area.

Reply: We have adapted the code and database to include the flag with the area of each pixel. Also, we provide a new option to generate ready-to-use WRFCHEM input files in hourly basis.

Reply to minor comments

There are many paragraphs consisting in less than two phrases. Each paragraph should have at leat three parts, intro, body and conclusion. Please fix.

English needs to be revised

Line 83: "The software provides…" Improve English.

Reply: We have fully revised the English in the manuscript.

Line 84: Given that there are fuel consumption data by month available in Brazil, why the authors provide monthly emissions instead of calculating annual emissions with monthly profiles? Are you planning to improve this database?

Reply: Yes, indeed. We have been planning to generate the database using monthly fuel consumption. However, the National Petroleum Agency has not provided this data until now.

Figure 2: Include the resolution in the figure, also, increase the size of the legend and fire itself.

Reply: Ok! We will fix the resolution and size from figure 2.

Lines 120-121: can you a plot of the emission factors of RCHO and CO for heavy-duty and light-duty vehicles?

Reply: We will provide this figure as supplementary material.

Figure 4: present the first three plots horizontally and below the expanded hourly temporal factors.

Reply: Ok! We will plot them horizontally and enlarge this figure.

Lines 181-184; I think the author is being ambiguous. One thing is having the emission files according CB6 and another is that the files are according to these models. Each one of these models have emissions input files with different NetCDF characteristics. Then the authors must be more careful choosing the right words.

Reply: We have fixed this issue in the manuscript. We have targeted the species from CB06 to elaborate the input files. Each file has a different NetCDF structure and flags.

Figure 8: Make figure bigger, reduce space between Brazil for each comparison, use more pages if needed.

Reply: Thanks. We have fixed figure 8 following your recommendations.

Figure 9: Correlation figures seems distorted. These figures could be a) EDGAR, b) BRAVES, c) bias with the same color legend. For instance, negative could be blue, positive red, centered at 0, with degradation.

Reply: We have fixed this issue in figure 9.

Line 217. Why EDGAR is higher than BRAVES? On a recent paper (Nogueira et al., 2021) it was found that the CETESB emission actors need to be corrected to represent tunnel emission factors. The correction based on this publication is shown below. This correction is already available in the VEIN model (Ibarra-Espinosa et al., 2018).

Reply: EDGAR is higher than BRAVES because most of the road transport EFs are overestimated in EDGAR, as reported by Madrazo et al. (2018). Also, large discrepancies were found by Huneeus et al. (2020) between EDGAR and local/national city emissions data

for the same domain. In addition, Huneeus et al. (2020) showed that transportation emissions from EDGAR were higher than local city inventory in Rio de Janeiro.

Madrazo, J., Clappier, A., Belalcazar L. C., Cuesta, O., Contreras H., Golay F., 2018. Screening differences between a local inventory and the Emissions Database for Global Atmospheric Research (EDGAR). Science Total Environment, 631–632, pp. 934-941. 10.1016/j.scitotenv.2018.03.094

Huneeus, N., Denier van der Gon, H., Castesana, P., Menares, C., Granier, C., Granier, L., Alonso, M., de Fatima Andrade, M., Dawidowski, L., Gallardo, L., Gomez D., Klimont , Z., Janssens-Maenhout, G., Osses, M., Puliafito, S. E., Rojas, N., Ccoyllo , O. S., Tolvett, S., Ynoue, R. Y., 2020. Evaluation of anthropogenic air pollutant emission inventories for South America at national and city scale. Atmospheric Environment, 235, 117606. 10.1016/j.atmosenv.2020.117606

Line 238: registered or circulating fleet?

Reply: Circulating fleet. We have included this information in the manuscript.

**Reply to comments by Reviewer #2:**

1. there are local estimations regarding the composition and speciation of particles and VOCs which could be considered in the model or at least discussed why they were not considered.

Reply: We have adapted the code and database to include local data for better representing the Brazilian reality.

2. The legend in Figures 5 and 6 should include that the speciation of VOC and PM is based on SPECIATE (from EPA).

Reply: Thank you. We have included "Speciation factors from US EPA Speciate" in figures 5 and 6 captions.

3. The authors need to discuss the role of the biofuels used in Brazil in the total emission of VOC and the speciation considering this fleet.

Reply: We have included a new discussion in the manuscript to provide a baseline to understand the biofuels in Brazil and the role of ethanol.

4. the segregation of the streets is considering the type of vehicles or only an average of the composition with homogeneous distribution.

Reply: We have calculated the road density considering all roads in the OpenStreetMaps shapefile. We have not segregated by fleet category or road type.

5. the graphical presentation of the difference between inventories is a good visual form but I suggest a table with the total emissions comparing the different inventories, including the Brazilian one.

Reply: Ok! We will provide this table as supplementary material.

---

## Author Response (AR1)

**Multispecies and high spatiotemporal resolution database of vehicular emissions in Brazil**

MS No.: essd-2022-74

Dear reviewers,

We greatly appreciate the comments and suggestions you provided, which are very constructive and have contributed to enhance the content of the revised manuscript. Please find below the reply to all the reviewer's comments.

Best regards

Leonardo Hoinaski and coauthors

**Reply to comments by Reviewer #1, Dr. Sergio Ibarra:**

Reply to major comments

It is important to check the validity of comparison of the emissions with MERRA. The thing is that, yes, it is possible to compare emissions with concentrations but under specific conditions, for instance, when the boundary layer is low. In this way, the air pollutant concentrations should be representative of the emissions, as shown by Gallardo et al., 2012. As the database provided by Hoinaski et al present hourly factors, my recommendation is to compare under a similar set of conditions such as the ones presented by Gallardo et al. Author can find similar research on literature. Regarding the air pollutant concentrations, authors could use the CAMS global reanalysis (EAC4) Copernicus https://www.ecmwf.int/en/forecasts/dataset/cams-global-reanalysis.

Reply: We appreciate your suggestions. We have improved this discussion and reinforced the limitations when using this approach. Indeed, a comparison between datasets in specific regions and hourly averaging time would be more accurate. We have included a comparison according to your suggestion. To compare EDGAR, BRAVES database, and MERRA on an hourly basis, we have multiplied both BRAVES database and EDGAR annual emissions by a temporal disaggregation factor. We will analyze these three databases on 01/01/2015 at 8:00, when the boundary layer is not fully developed, and the traffic has the first peak period.

We have included the following sentences and figure in the manuscript:

"*The BRAVES database and EDGAR reach a similar spatial correlation with MERRA-2 when using annual averages (Figure SM8). The zoom-in quadrant in the São Paulo metropolitan region in Figure 10 reveals a greater level of details from the BRAVES database compared to EDGAR. In addition, BRAVES has higher temporal resolution and chemical speciated emissions and has presented a better correlation with MERRA-2 when comparing hourly averages (Figure 10). In Figure 10, we have compared MERRA-2 and emissions on 01/01/2015 at 8:00, when the boundary layer is low and the concentrations are representative of the emissions, as shown by Gallardo et al. (2012). It is worth emphasizing that the straightforward comparison between emission and concentrations from monitors or reanalysis data must be made carefully and under specific conditions.*"

[Figure]

*Figure 10. CO spatial distribution provided by (a) MERRA2, (b) EDGAR, and (c) BRAVES. Scatter plots of CO vehicle emission and CO hourly surface concentrations in SP on 01/01/2015 at 8:00.*

Lines 55-58: "Current inventories provide only annual emissions not reaching the spatial … resolution… nor the concentration of chemical species…" According the to these words, the authors are stating the one problem of the emissions inventories are not providing concentrations. This is conceptually wrong because emissions are mass and concentrations mass over volume. I think the author meant the inventories usually do not provide the required speciation, which would be an English problem, but need to check.

Reply: Yes, you are right. We have fixed this issue in the manuscript by replacing "nor the concentration of chemical species" with "nor the emission of chemical species".

Line 85: According to the CETESB (2019, the same reference used by author), RCHO is actually the sum of aldehydes and formaldehyde. Furthermore, CETESB also provides NMHC - ETOH emission factors. In addition, as the road transportation fuel consumed in Brazil has a vastly origin on bio-fuels, 27% of gasoline is ethanol and 7% of diesel is bio-diesel, this results in a unique chemical signature of the chemical composition of fuel, hence the emissions. Actually, there are literature mentioning the high number of carbonyls in the vehicular emissions (Nogueira et al., 2015). According to table SM7, the species C2H4O (ALD2), C2H6O (ETOH) and CH2O (FORM PRIMARY) are present in the NMHC speciation. Then, in order to provide a correct speciation, the author must reply the following questions:

Which pollutant are they using to perform the speciation? NMHC or NMHC – ETOH? Do the NMHC emission factors already consider RCHO? If the authors are using NMHC, the it supposed that ETOH and RCHO is already part of NMHC. However, this would result in a different chemical composition signature. Then, the recommendation would be preserving the proportions of ALD, FORM and ETOH and recalculate the speciation for the other compounds. I would say that this is more important for ETOH than RCHO. Can the author comment on that? Is this part of an ongoing work?

Reply: Thank you for this very insightful comment. We have fully adapted the code to preserve the original estimates using CETESB's emission factors, which is the best way to represent the ETOH and RCHO emissions in Brazil.

Now, we speciate the acetaldehydes (ALD2 and ALD2_PRIMARY), formaldehyde (FORM and FORM PRIMARY), and aldehydes with 3 or more carbons (ALDX) using RCHO estimates, which is exclusively estimated from local emission factor in BRAVES. We have considered that acetaldehyde represent 50% of RCHO emissions from light-duty vehicles. Formaldehyde emissions (FORM and FORM_PRIMARY) represents 39% of RCHO emissions from light-duty. ALDX has been considered as 10% of RCHO emissions, while acetone (ACET) accounts for 8% of these emissions.

We have also modified the code and database to include the ETOH emissions from Flex Fuel vehicles running with ethanol when CETESB's emission factor is available (since 2018). ETOH emissions from other vehicles will still be estimated using NMHC speciation factors from Speciate.

Speciation factors and figures used in this work have been fully reviewed. See table at https://github.com/leohoinaski/BRAVES/blob/main/ChemicalSpec/BRAVES_speciation.csv and figure bellow. We have also included the following sentences:

*"We speciate the acetaldehydes (ALD2 and ALD2_PRIMARY), formaldehyde (FORM and FORM PRIMARY), and aldehydes with 3 or more carbons (ALDX) using original RCHO estimates from BRAVES, which are based on local emission factor from Companhia de Tecnologia de Saneamento Ambiental do Estado de São Paulo - CETESB (CETESB, 2022). We have considered that acetaldehyde represents 50% of RCHO emissions from light-duty vehicles. Formaldehyde emissions (FORM and FORM_PRIMARY) represent 39% of RCHO emissions from light-duty (Nogueira et al. 2015). ALDX has been considered as 10% of RCHO emissions, while acetone (ACET) accounts for 8% of these emissions.*

*We have also kept the estimated using local emissions factor for ethanol (ETOH), which has been the best way to represent the particularities of biofuels in Brazil. CETESB has provided the ETHO emission factors since 2008, for light-duty vehicles running with ethanol and gasoline (CETESB, 2022). Since CETESBs' RCHO and ETOH emissions factors are available only for light-duty and commercial light vehicles, we have used percentage factors from Speciate to estimate aldehyde and ethanol emissions from NMHC for motorcycles and heavy-duties.*

*The Brazilian gasoline C, which has fueled light-duty vehicles, is a mixture of pure gasoline and 20 to 25% of anhydrous ethanol. Since 2008, heavy-duty vehicles have run with a blend of diesel and up to 15% of biodiesel. This unique chemical signature of the biofuels in Brazil reflected significantly in the vehicular emissions, especially those of carbonyls and ethanol (Nogueira et al. 2015; CNPE, 2018). These last compounds deserve attention since they are major precursors of tropospheric ozone (Atkinson, 2000, Jacob, 2000)."*

[Figure]

[Figure]

Lines 102-104: While this method is conservative, the emissions need to be considered are currently expressed as points in space and not mass flux. Then, the current format is not appropriate for air quality models. In other words, the temporal mass must be divided by the area, for instance, the gases inputs to the WRF Chem model are expressed as $\mu g/km^2/h$. Then, the authors must recalculate the NetCDF outptus or add the proper flag indicating that the user must do divide by the area.

Reply: We have adapted the code and database to include the flag with the area of each pixel. Also, we provide a new option to generate ready-to-use WRFCHEM input files in hourly basis. The following sentence has been added in the manuscript:

*"Flags have been included in the netCDF files to provide the area and time zones of each pixel, so users can choose the option to generate ready-to-use hourly input files for CMAQ (in mass or mol per second) or WRFCHEM (in mass or mol flux per area)."*

Reply to minor comments

There are many paragraphs consisting in less than two phrases. Each paragraph should have at leat three parts, intro, body and conclusion. Please fix.

English needs to be revised

Line 83: "The software provides…" Improve English.

Reply: We have fully revised the English in the manuscript.

Line 84: Given that there are fuel consumption data by month available in Brazil, why the authors provide monthly emissions instead of calculating annual emissions with monthly profiles? Are you planning to improve this database?

Reply: Yes, indeed. We have been planning to generate the database using monthly fuel consumption. However, the National Petroleum Agency has not provided this data until now. We have added the following sentence in the conclusion chapter:

*"Temporal variability would also be improved by regionalizing the profiles to account for the traffic flow in each location or by including monthly fuel consumption data."*

Figure 2: Include the resolution in the figure, also, increase the size of the legend and fire itself.

Reply: Ok! We included the spatial resolution in the description of the figure 2. Also, we increased the size of the legend.

[Figure]

*Figure 2. Road density factor in Brazil with spatial resolution of 0.05°×0.05°.*

Lines 120-121: can you a plot of the emission factors of RCHO and CO for heavy-duty and light-duty vehicles?

Reply: Ok! We have plotted the figure with the emission factors of these pollutants. For the RCHO, CETESB provides the emission factors only for light-duty and light-duty commercial vehicles. RCHO emitted by heavy-duty and motorcycles have been estimated by speciation factor from SPECIATE.

BRAVES estimates weighted average emission factors by municipality, which includes the influence of fleet characteristics. We have included scrappage and deterioration factors. You can see the weighted emission factor for RCHO and CO below.

[Figure]

Figure 4: present the first three plots horizontally and below the expanded hourly temporal factors.

Reply: We believe that the original file has enough quality, and it has lost quality while generating the pdf file. We have been planning to plot this figure in a single column, as in this example:

sizes are proportional to the sizes of the catchments. The grey line in (c) and (d) indicates the limits of hydrographic regions.

[Figure]

Figure 2. Time series with the number of streamflow gauges with at least one measurement for a given year in Brazil.

meteorological variable with the highest spatial heterogeneity amongst those used in CAMELS-BR.

In addition to GLEAM v3.3a, estimates of actual evapotranspiration (ET) were obtained from the MGB model version for South America (Siqueira et al., 2018). The MGB is a conceptual, semi-distributed hydrologic–hydrodynamic model that discretizes the basin (or a set of basins) into irregular unit catchments and further into hydrological response units by combinations of land use and soil types, where both water and energy balance are computed. The model calculates ET using the Penman–Monteith equation based on CRU meteorological data (i.e., temperature, pressure, radiation, and wind speed) and MSWEP v1.1 precipitation data (Beck et al., 2017b). Surface resistance is adjusted according to the availability of water in the soil that is updated during the water budget. The MGB also computes the evaporation of flooded areas and intercepted water from the canopy with the Penman equation. Regular ET cells of 0.5° resolution were generated by aggregating unit catchments using their areas as weights.

The long-term water balance is accurate for most catchments, using either the estimated evapotranspiration from GLEAM (Fig. 3a) or MGB (Fig. 3b). Both evapotranspiration data sources indicate that the highest data uncertainties occur in the Amazon and smaller catchments in the Paraná and the southeastern Atlantic regions since those catchments are further away from the 1 : 1 line in Fig. 3a–b. The same conclusions are derived from visualizing the runoff coefficient as a function of the humidity index (Fig. 3c). In addition, there are remarkable differences between GLEAM and MGB estimates, where evapotranspiration from GLEAM is substantially higher in the Amazon basin and substantially lower in the eastern and the western Northeast Atlantic regions.

https://doi.org/10.5194/essd-12-2075-2020

Earth Syst. Sci. Data, 12, 2075–2096, 2020

We prefer to keep this figure as it is:

[Figure]

Lines 181-184; I think the author is being ambiguous. One thing is having the emission files according CB6 and another is that the files are according to these models. Each one of these models have emissions input files with different NetCDF characteristics. Then the authors must be more careful choosing the right words.

Reply: We have fixed this issue in the manuscript. We have removed all references to CB06, since our estimates could be used in other chemical mechanisms.

Figure 8: Make figure bigger, reduce space between Brazil for each comparison, use more pages if needed.

Reply: Thanks. We have fixed figure 8 following your recommendations.

EDGAR                                    BRAVES

[Figure]

Figure 9: Correlation figures seems distorted. These figures could be a) EDGAR, b) BRAVES, c) bias with the same color legend. For instance, negative could be blue, positive red, centered at 0, with degradation.

Reply: We have fixed this issue in figure 9.

[Figure]

Line 217. Why EDGAR is higher than BRAVES? On a recent paper (Nogueira et al., 2021) it was found that the CETESB emission actors need to be corrected to represent tunnel emission factors. The correction based on this publication is shown below. This correction is already available in the VEIN model (Ibarra-Espinosa et al., 2018).

Reply: This pattern has been reported by Huneeus et al. (2020), Álamos et al. (2022) and Madrazo et al. (2018). An evaluation using BRAVES database as input in air quality models would bring important information about the model's errors and representativeness.  The following sentence has been included in the manuscript:

*"An evaluation using BRAVES database as input in air quality models would bring important information about the model's errors and representativeness. As reported by Nogueira et al., (2021), the emission factors from CETESB used in this work would require future corrections to better represent field measurements."*

Line 238: registered or circulating fleet?

Reply: We have fixed this issue. We have added the following sentence in section 7 of the manuscript:

*"In 2021, ~31 million vehicles were registered in SP state, being considered the state with the highest vehicular emission in Brazil (SENATRAN 2021; Vasques and Hoinaski, 2021)."*

**Reply to comments by Reviewer #2:**

1. there are local estimations regarding the composition and speciation of particles and VOCs which could be considered in the model or at least discussed why they were not considered.

Reply: We have adapted the code and database to include local data for better representing the Brazilian reality.

*"We speciate the acetaldehydes (ALD2 and ALD2_PRIMARY), formaldehyde (FORM and FORM PRIMARY), and aldehydes with 3 or more carbons (ALDX) using original RCHO estimates from BRAVES, which are based on local emission factor from Companhia de Tecnologia de Saneamento Ambiental do Estado de São Paulo - CETESB (CETESB, 2022). We have considered that acetaldehyde represents 50% of RCHO emissions from light-duty vehicles. Formaldehyde emissions (FORM and FORM_PRIMARY) represent 39% of RCHO emissions from light-duty (Nogueira et al. 2015). ALDX has been considered as 10% of RCHO emissions, while acetone (ACET) accounts for 8% of these emissions.*

*We have also kept the estimated using local emissions factor for ethanol (ETOH), which has been the best way to represent the particularities of biofuels in Brazil. CETESB has provided the ETHO emission factors since 2008, for light-duty vehicles running with ethanol and gasoline (CETESB, 2022). Since CETESBs' RCHO and ETOH emissions factors are available only for light-duty and commercial light vehicles, we have used percentage factors from Speciate to estimate aldehyde and ethanol emissions from NMHC for motorcycles and heavy-duties."*

2. The legend in Figures 5 and 6 should include that the speciation of VOC and PM is based on SPECIATE (from EPA).

Reply: Thank you. We have included "Speciation factors from US EPA Speciate" in figures 5 and 6 captions.

3. The authors need to discuss the role of the biofuels used in Brazil in the total emission of VOC and the speciation considering this fleet.

Reply: The following sentence has been added in the manuscript to introduce the usage of biofuels in Brazil:

*"The Brazilian gasoline C, which has fueled light-duty vehicles, is a mixture of pure gasoline and 20 to 25% of anhydrous ethanol. Since 2008, heavy-duty vehicles have run with a blend of diesel and up to 15% of biodiesel. This unique chemical signature of the biofuels in Brazil reflected significantly in the vehicular emissions, especially those of carbonyls and ethanol (Nogueira et al. 2015; CNPE, 2018). These last compounds deserve attention since they are major precursors of tropospheric ozone (Atkinson, 2000, Jacob, 2000)."*

4. the segregation of the streets is considering the type of vehicles or only an average of the composition with homogeneous distribution.

Reply: We have calculated the road density considering all roads in the OpenStreetMaps shapefile. We have not segregated by fleet category or road type.

5. the graphical presentation of the difference between inventories is a good visual form but I suggest a table with the total emissions comparing the different inventories, including the Brazilian one.

Reply: Ok! We provided this table as supplementary material. A full comparison between BRAVES emissions and others Brazilian inventories in different aggregated spatial scales is available in Vasques and Hoinaski (2021).

Vasques TV, Hoinaski L (2021). Brazilian vehicular emission inventory software – BRAVES. Transportation Research Part D: Transport and Environment 100:103041. https://doi.org/10.1016/j.trd.2021.103041

We have added the following sentence in section 7 of the revised manuscript:

*"Table SM7 also shows a comparison of the total vehicular emissions aggregate in Brazilian territory, considering BRAVES, EDGAR, and others available national inventories."*

*Table SM7. Total emission of CO, NOx, MP, and NMVOC aggregated in Brazilian territory from available national inventories.*

| National Inventory | Base Year | CO | NOx | MP | NMVOC |
|---|---|---|---|---|---|
| BRAVES | 2013 | 1433499 | 924752 | 30322 | 272430 |
| SEEGv8.0 | 2013 | 1337408 | 1196302 | - | 236108 |
| MMA 2013 | 2012 | 1251596 | 1100801 | 37231 | 250000 |
| EDGARv5.0 | 2013 | 9129167 | 1691271 | 89206 | 1650005 |